# “NO” Time in Fear Response: Possible Implication of Nitric-Oxide-Related Mechanisms in PTSD

**DOI:** 10.3390/molecules29010089

**Published:** 2023-12-22

**Authors:** Mariana G. Fronza, Bruna F. Ferreira, Isabela Pavan-Silva, Francisco S. Guimarães, Sabrina F. Lisboa

**Affiliations:** 1Pharmacology Departament, Ribeirão Preto Medical School, University of São Paulo, São Paulo 14049-900, Brazil; marianagfronza@usp.br (M.G.F.); brunaferreira@usp.br (B.F.F.); isabelapavanns@usp.br (I.P.-S.); 2Biomolecular Sciences Department, School of Pharmaceutical Sciences of Ribeirão Preto, University of São Paulo, São Paulo 14040-903, Brazil

**Keywords:** nNOS, post-translational modifications, memory

## Abstract

Post-traumatic stress disorder (PTSD) is a psychiatric condition characterized by persistent fear responses and altered neurotransmitter functioning due to traumatic experiences. Stress predominantly affects glutamate, a neurotransmitter crucial for synaptic plasticity and memory formation. Activation of the N-Methyl-D-Aspartate glutamate receptors (NMDAR) can trigger the formation of a complex comprising postsynaptic density protein-95 (PSD95), the neuronal nitric oxide synthase (nNOS), and its adaptor protein (NOS1AP). This complex is pivotal in activating nNOS and nitric oxide (NO) production, which, in turn, activates downstream pathways that modulate neuronal signaling, including synaptic plasticity/transmission, inflammation, and cell death. The involvement of nNOS and NOS1AP in the susceptibility of PTSD and its comorbidities has been widely shown. Therefore, understanding the interplay between stress, fear, and NO is essential for comprehending the maintenance and progression of PTSD, since NO is involved in fear acquisition and extinction processes. Moreover, NO induces post-translational modifications (PTMs), including S-nitrosylation and nitration, which alter protein function and structure for intracellular signaling. Although evidence suggests that NO influences synaptic plasticity and memory processing, the specific role of PTMs in the pathophysiology of PTSD remains unclear. This review highlights pathways modulated by NO that could be relevant to stress and PTSD.

## 1. Introduction

Fear and stress responses are adaptive reactions of the organism that enable coping with environmental challenges and threats. These responses encompass cognitive, emotional, and physiological changes, with the brain playing a pivotal role in their regulation [1,2]. The manifestation of exaggerated and abnormal fear responses, particularly following traumatic experiences, represents a distinct behavioral phenotype within the spectrum of PTSD. This behavioral profile is characterized by intense feelings of fear, horror, or helplessness. Additionally, PTSD is marked by the pervasive presence of intrusive memories, avoidance behaviors and persistent negative changes in thinking and mood [3,4]. Preclinical and clinical studies have shown alterations in the circuit that connects the amygdala, hippocampus, and medial prefrontal cortex (mPFC) in PTSD subjects, which also underlies fear behavior in mammals [5,6]. These brain regions are involved in the regulation of emotions and responses to stressful situations [7,8].

NO is a gaseous neuromodulator that regulates, among several other functions, neuronal function, particularly related to learning and memory [9], and fear-related and defensive responses [10]; it thereby plays an important role in related psychiatric disorders [11], including in PTSD [12]. For example, polymorphisms on NO pathway genes, NOS1 (nNOS) and its adaptor protein NOS1AP are involved in PTSD severity, depression, anxiety, stress, and resilience [13,14,15], which strongly suggest its involvement in the pathophysiology of this disorder. Moreover, in animal models of PTSD, there are reports of changes in NO and nNOS in the brain [16,17]. Additionally, nNOS is present in several limbic system areas involved in the regulation of emotions and responses to stressful situations, including the amygdala, mPFC, periaqueductal gray matter (PAG), interstitial nucleus of stria terminalis hypothalamic paraventricular nucleus, and hippocampus [18].

nNOS represents the predominant NO-producing enzyme in the brain, and homozygous knockout (KO) mice for this isoform showed residual production of NO (>10%) [19] accounted for by the presence of endothelial NOS (eNOS, NOS3) in hippocampal pyramidal cells [20]. Moreover, inducible NOS (iNOS, NOS2) was considered to only be induced by inflammatory stimuli, it is now known that it is constitutively expressed at low levels in some brain regions [21,22,23] and can be induced by stress [24,25]. Therefore, iNOS also accounts for NO production in the brain, and its deletion or pharmacological inhibition can also modulate behavior, including causing antidepressant responses in mice [21] and attenuation of anxiety induced by stress [26], including with local inhibition in the mPFC [27]. Moreover, in an animal model of PTSD there was a long-term increased NOS activity in the hippocampus, which was attenuated by an iNOS inhibitor [28], suggesting this isoform could also be a target in PTSD.

In this review, we will discuss studies evaluating how NO is involved in learned fear. Moreover, we will summarize biological alterations involving NO that might play a role in the neurobiology of PTSD, with a focus on PTMs (Figure 1). One of the questions that motivate this review is to identify if nNOS underlying pathways exert a significant role in response to trauma and in the pathophysiology of fear-related disorders.

### NO Production and Signaling Pathways

NO is a diatomic free radical with a unique chemistry among other small molecule signaling agents. This molecule is endogenously synthesized in cells through the conversion of L-arginine into L-citrulline by NOS enzymes [29]. This complex reaction involves two independent steps intermediated by the cofactor NADPH. First, the hydroxylation of the guanidino nitrogen in arginine forms the intermediate product Nω-hydroxy-L-arginine, which is sequentially oxygenated in a reaction catalyzed by the tetrahydrobiopterin cofactor, resulting in NO and L-citrulline [30,31]. NOS isoforms can be differentiated by their tissue constitutive expression (nNOS and eNOS) or by their functional regulatory mode (iNOS); they can also be classified based on calcium (Ca^2+^)-dependency (nNOS, eNOS) or independency (iNOS) [32].

NO plays a key role in the regulation of neurotransmission, cardiovascular function, immunity, and other biological functions [29,33,34]. In the brain, NO is considered an atypical neurotransmitter, mainly produced in postsynaptic neurons and traveling through the extracellular space to act directly in the presynaptic neuron, inducing long-term potentiation (LTP, a form of neuronal plasticity) and also coordinating the release of neurotransmitters (for more detailed information, topic 9) [35,36,37]. NO also exerts direct effects on postsynaptic neurons [38,39], which will be discussed further.

Considering the gaseous form of NO, once it is released from its attachment to Fe^2+^/Fe^3+^ ion within the heme group of NOS, it may readily diffuse over cell membranes exerting effects mediated by canonical, non-canonical, and less canonical pathways. In the canonical pathway, NO selectively activates the soluble guanylate cyclase (sGC) enzyme, majorly located at the presynaptic terminal [40], through rupture of the His–Fe^2+^ bond within the heme group of sGC. This NO action promotes a conformational change resulting in increased conversion of guanosine-5′-triphosphate (GTP) to cyclic guanosine monophosphate (cGMP) and therefore the activation of cGMP-activated targets (serine/threonine protein kinases-cGKs or PKGs, channels, and phosphodiesterases), which are for instance required for LTP induction. In contrast, in the less canonical pathway NO competes with oxygen (O_2_) in cytochrome c oxidase (the terminal enzyme of the mitochondrial electron transport chain), considered a negative physiological regulator of respiration in tissue. In this pathway, NO-induced blockade of mitochondrial respiration can result in the excitotoxic death of neurons due to the induced release of glutamate and activation of NMDA-type glutamate receptors [41]. Finally, in the non-canonical pathway, NO affects mainly cysteine (thiol groups) and tyrosine residues in proteins through covalent PTMs, such as nitrosylation and nitration, affecting protein folding and activity [42,43,44].

The existing body of literature currently covers approximately 3000 S-nitrosoproteins, including those discovered through the external administration of nitrosylating agents and physiological processes [45]. Among them, the most well-elucidated NO-induced PTMs include S-nitrosylation and tyrosine nitration, which are mainly differentiated by their capacity to be reversible. Tyrosine nitration is a process mediated by peroxynitrite (ONOO^−^) produced by the reaction of NO and the free radical anion superoxide (O^2−^), directly related to oxidative stress. Peroxynitrite adds a nitro group to the 3-position adjacent to the hydroxyl group of tyrosine to produce the stable product 3-nitro-tyrosine, an index of peroxynitrite-mediated nitration [46]. Thus, the incorporation of a nitro group to tyrosine irreversibly modifies key properties of this amino acid, including phenol group pKa, redox potential, hydrophobicity, and volume, thus altering the structure and function of the given protein [47,48].

Regarding S-nitrosylation, this is a conserved mechanism between species for regulatory and effector modifications, that may ultimately lead to the formation of disulfide bonds through the reaction of NO with a cysteine thiol of a protein [49]. This modification can regulate a wide range of events including subcellular localization, protein-protein interactions, and protein stability. Moreover, it can be reversed by denitrosylases using reducing equivalents derived from hydroxylamine reductase (NADH) or NADPH by two distinct systems: thioredoxin (Trx)/TR and the glutathione (GSH)/S-nitrosoglutathione (GSNO) reductase (GSNOR) systems [50]. S-nitrosylation can also trigger a transnitrosylation cascade leading to the regulation of intracellular pathways. In this sense, the S-nitrosylation consists firstly of the transition of NO (a radical with an unpaired electron occupying an antibonding π orbital) that after reacting with a cysteine thiol of a protein, loses the unpaired electron becoming NO^+^ (nitrosonium cation, oxidized NO) [51]. Thus, for the transnitrosylation reaction to be effective, it has to be catalyzed by either metal transition (Cu, Zn, Fe) or by the enzymatic reaction with NADH and nitrate reductase, at least in anaerobic conditions [52]. Therefore, in the transnitrosylation reaction, the transfer of an NO^+^ group (S-nitroso functional group) to the cysteine thiol in the side chain of another protein (R group) involves this thiolate anion acting as a nucleophile and performing a reversible nucleophilic attack on the nitroso nitrogen to form an SNO-protein adduct. Since the products of this reaction are an S-nitrosothiol and a thiol, a similar reaction in reverse will yield the original reactants [53,54]. This process can also be mediated by either a small SNO-modified molecule or a proximal SNO-modified protein acting as a transnitrosylase.

Over the past decades, increasing evidence indicates that S-nitrosylation regulates other PTMs, including phosphorylation, acetylation and ubiquitylation (for more details see [55]). For example, S-nitrosylation appears to directly compete with cysteine sites for S-palmitoylation, a PTM which involves the attachment of palmitoyl moieties to specific cysteine residues through thioester bonds [56,57]. This PTM leads to increased hydrophobicity and enhances the affinity for plasma membranes [58]. S-nitrosylation might also regulate alternative redox modifications, such as glutathionylation or sulfhydration, through the prevention of further oxidation of protein thiols [59,60].

## 2. NO Involvement in Learned Fear

Understanding mechanisms related to the symptoms of PTSD and the normal fear response often relies on studying learned fear and evaluating conditioned responses in animal models [61]. In this sense, it was observed that nNOS KO mice showed a significant impairment in contextual and cued fear learning when compared with wild-type (WT) mice [62]. Additionally, the administration of the NO donor molsidomine to nNOS KO mice improved the previous deficits in short- and long-term contextual fear memory [62]. Interestingly, when nNOS+ interneurons were transplanted in the dentate gyrus of nNOS KO mice, the acquisition of fear memory was restored, indicating the necessity of nNOS+ neurons in learning, in this particular case of an aversive memory [63]. Conversely, iNOS KO mice, which might have a compensatory increase in nNOS activity in the brain, besides not presenting differences in contextual fear conditioning expression, have deficits in fear extinction when compared to WT animals [64]. Therefore, there is a fine-tuned control of learned fear by NO, particularly nNOS, indicating that although NO production is necessary for learning, exaggerated NO production can contribute to aversive responses and persistence of fear.

Pharmacological inhibitors of nNOS have shown interesting findings in the literature regarding the enzyme’s role in fear responses. Despite evidence demonstrating that systemic administration of an nNOS inhibitor, 7-Nitroindazole (7-NI), did not affect the acquisition of contextual fear [65], recent studies have shown the opposite. For instance, Kelley et al. (2010) demonstrated that administration of the selective nNOS inhibitor, S-methyl-l-thiocitrulline, to WT mice before fear conditioning impaired both short- and long-term memories of contextual but not cued conditioned fear [66]. Moreover, using a two-process contextual fear conditioning paradigm and 7-NI, it was suggested that nNOS contributes to the acquisition but not consolidation of context-shock associative memory [67]. These data suggest that NO produced by nNOS may have a primary role in learned fear, modulating its acquisition.

Regarding fear expression, freezing behavior during expression of contextual fear conditioning and NO levels in the dorsal hippocampus of rats were positively correlated [68], indicating the importance of this atypical neurotransmitter, at least in the hippocampus, to the proper expression of a fear response. In fact, the administration of nNOS inhibitors in fear-related brain areas, including the dorsal hippocampus [68] and the mPFC [69] attenuated the expression of learned conditioned fear.

nNOS is also important for fear extinction. Administration of the nNOS inhibitor ZL006, a drug that uncouples nNOS from PSD95, into the CA3 of the hippocampus facilitates fear extinction [70], and its administration into the dentate gyrus promotes extinction retrieval of remote fear extinction [71]. Corroborating this data, the combination of MK-801 (NMDA antagonist) and L-NNA (NOS inhibitor) treatment ameliorated the extinction learning deficits in mice exposed to chronic social isolation stress [72]. Moreover, iNOS KO mice, which present with increased NOS activity in the prefrontal cortex and altered expression of nNOS and eNOS enzymes, also have deficits in fear extinction which were attenuated by systemic 7-NI [64]. Interestingly, ZL006 in the anterior cingulate cortex (ACC) also inhibited contextual fear generalization in a novel context [73], another feature of PTSD.

Moreover, administration of 3-bromo-7-indazole, a selective NO synthesis inhibitor, and a blocker of protein de novo synthesis, cycloheximide, during memory reactivation prevented conditioned reflex freezing reactions. This suggests a potential role of NO in the labilization of contextual fear memories [74].

Fear renewal also seems to require NO-related mechanisms. This process is inhibited when 7-NI or ZL006 are injected before fear renewal in the lateral amygdala (LA) [75]. Moreover, scavenging reactive oxygen species (ROS) with N-acetyl cysteine, a potent antioxidant, in LA before extinction training attenuates fear renewal. These drugs indirectly hinder protein S-nitrosylation, indicating involvement of this process for fear renewal, at least in LA [75,76]. Conversely, L-NG-Nitro arginine methyl ester (l-NAME), a non-selective nNOS inhibitor, produced a dose-dependent impairment of fear extinction, in an ABB design of a three-tone fear extinction task in rats [77].

Time-dependent sensitization (TDS) stress, a PTSD model, increased hippocampal levels of NO metabolites, nitrogen oxides (NOx), on day 7 post-stress, which was blocked by 7-NINA, a non-selective NOS inhibitor administered immediately after stress [16].

Overall, these data indicate that inhibiting nNOS in the brain could ameliorate several aspects of fear memory, with implications for PTSD.

### Molecular Pathways Modified by nNOS and Implications for Fear Response

The human *nNOS* gene (*NOS1*) is located at chromosome 12q24.2, consisting of 29 exons and 28 introns, interspersed over a 250 Kb genome (Figure 2). Interestingly, the expression of the *nNOS* gene is regulated by many cellular processes, including by the transcription factor activator protein 2 (AP-2), T-cell factor/lymphoid enhancer factor 1 (TCF/LEF1), cAMP-response element binding protein (CREB)/activating transcriptional factor (ATF)/c-Fos, erythroblast transformation specific (Ets), p53, and nuclear factor kappa B (NF-kB) [78]. *nNOS* has various splicing variants and different subcellular localizations. nNOSa is the predominant splice variant in the brain and contains an N-terminal PSD/Discs-large/ZO-1 homologous (PDZ)-binding domain containing 82 residues. The PDZ domain determines the cellular distribution of nNOS, targeting this protein to post or presynaptic sites mediated by its adaptor protein NOS1AP (initially termed CAPON) [79,80]. In postsynaptic neurons, the nNOS PDZ domain interacts with other PDZ domains located on PSD95 and PSD93 proteins. Among the three different PDZ motifs of PSD95, the first one interacts with the glutamatergic receptor NMDAR (NR2a or b subunit) whereas the third PDZ interacts with skeletal proteins responsible for anchoring [81]. This protein complex consisting of NMDAR/PSD95/nNOS allows the efficient production of NO by activating NMDAR and is shown to be modulated or adapted by NOS1AP [82,83,84].

The PSD95/NOS1AP/nNOS complex within the amygdala mediates synaptic strengthening and is essential for fear consolidation [86]. As mentioned before, its disruption regulates fear extinction in the dorsal CA3 region of the hippocampus through the facilitation of brain-derived neurotrophic factor (BDNF)/Tropomyosin-related kinase receptor B (TrkB) signaling [70]. Moreover, in the anterior cingulate cortex, the inhibition of the PSD95/NOS1AP/nNOS complex prevented fear generalization through the reduction of histone deacetylase 2 (HDAC2) enzyme expression [73]. HDAC2 is one of the main players involved in remote fear memory formation through chromatin modification, promoting stable and long-lasting effects on the transcriptional regulation of gene expression [87]. These findings support the hypothesis that disrupting nNOS and its underlying pathways might be beneficial for fear-related disorders, such as PTSD, which could involve chromatin remodeling.

Extinction memory in the infralimbic (IL) subregion of mPFC relies on NMDAR-induced extracellular regulated kinases (ERK) signaling [88]. NMDAR activation leads to nNOS binding with NOS1AP, causing ERK dysfunction via Dexras1 S-nitrosylation [89]. Extinction training elevates nNOS-NOS1AP association in the IL, and the binding of NOS1AP and nNOS hinders extinction memory. Disrupting nNOS-NOS1AP can prevent the re-emergence of extinguished fear in mice [90]; it also enhances ERK phosphorylation, supporting memory retention in an ERK2-dependent manner, promoting synaptic changes, spine density, and preventing fear recovery in mice [90].

nNOS might also be located in the presynaptic sites of neuronal cells through a quaternary complex via interaction with the phosphotyrosine (PTB) domain of the N-terminus of the NOS1AP protein sequence, and synapsins I, II and III [79]. The NOS1AP PTB domain is also responsible for the interaction of nNOS/NOS1AP with other molecules, including DexRas-1 [91] and Scribble [92], which are essential for the propagation of NO signaling.

Additionally, several nNOS partners have been functionally involved in endogenous mechanisms regulating NO synthesis including nNOS-inhibiting protein (PIN), nitric oxide-interacting protein (NOSIP) asymmetric dimethylarginine (ADMA) and dimethylarginine dimethylaminohydrolases (DDAH). PIN is a well-conserved 10 kDa peptide that interacts with 163–245 residues of nNOS destabilizing the nNOS dimer, and therefore the conformation necessary for its activity [93,94]. Similarly, NOSIP, an E3 ubiquitin-protein ligase of 35 kDa has also shown the ability to reduce nNOS/eNOS activity as well as sub-cellular distribution, ultimately regulating processes such as neuronal development, apoptosis, and cell proliferation [95]. Dimethylarginine dimethylaminohydrolases (DDAH) are a family of enzymes that metabolize methylated arginines. Two isoforms, DDAH1 and DDAH2, give rise to four forms of methylated arginine. The former is expressed in the amygdala, striatum, cerebellum, thalamus, hippocampus and various cortical areas, while the latter has a very low expression in the brain, mostly restricted to the medulla and spinal cord [76]. A potent inhibitor of NOS activity is the isoform Nω,Nω′-dimethyl-l-arginine (asymmetric dimethylarginine, ADMA) [95], which causes uncoupling of NOS, a process that leads to the production of superoxides [96]. DDAH inhibition results in increased ADMA concentrations and reduced NO synthesis. DDAH1 is primarily co-expressed with *nNOS* in the brain, and *DDAH2* is detected in tissues that also express *eNOS* and *iNOS* [76].

These data reinforce the multiple roles underlying nNOS signaling and its importance in regulating neurotransmission in the brain. Therefore, nNOS and its underlying pathways could exert a significant role in response to trauma and in the pathophysiology of fear-related disorders. In the upcoming discussion, we will explore the potential impact of NO and NO-induced PTMs on the impaired aspects of fear memory related to PTSD physiopathology. These aspects include stress response, neuroinflammation, oxidative stress, cell death, and neurotransmission/synaptic plasticity.

## 3. nNOS, NO-Related Mechanisms, and Stress Response

While the relationship between NO and the regulation of the stress response were once considered contradictory [97], it is now widely accepted that nNOS plays a significant role in modulating the hypothalamic-pituitary-adrenal (HPA) axis and the emotional responses following episodes of stress (for more information, see Table 1). For instance, direct injection of NO into the brain has been demonstrated to stimulate the activity and transcription of genes encoding nerve growth factor-induced B (NGFI-B) and corticotropin-releasing factor receptor type 1 (CRHR1) in the paraventricular nucleus of the hypothalamus and the central nucleus of the amygdala (CeA), ultimately leading to increased adrenocorticotropic hormone (ACTH) plasma levels [98]. Meanwhile, intracerebroventricular (i.c.v.) administration of L-NAME abolished the plasma ACTH response to intermittent electroshock and concurrently decreased hypothalamic NOS activity [99].

Regarding the involvement of NOS, particularly nNOS, in stress response, the upregulation of nNOS has been observed in key cerebral areas involved in emotional responses in both acute and chronic stress paradigms, although in a structure- and time-dependent fashion. For instance, a brief 2 h-episode of acute restraint stress, modeling a trauma, led to a substantial rise in the density of neurons expressing nNOS in the amygdaloid nucleus after 2 h. However, a similar increase in neuronal density in the hippocampus and entorhinal cortex required nearly 5 days of exposure to the stressor [106]. Additionally, rapid upregulation of *nNOS* gene expression and its activity in the PFC, but not in the striatum, were noted after 60 min of restraint stress, with subsequent upregulation at 240 min [107,108]. Moreover, specifically in the prelimbic (PL) portion of mPFC, a single 3 h-restraint stress session induced increased nNOS expression after 24 h or 7 days in this brain region [100]. Finally, exposure of rats to a live cat, a model of trauma also increased nNOS expression and NOx levels, an indirect measure of NOS activity, in the mPFC one week later, which correlated with freezing behavior during the stress exposure [17].

Corroborating a possible causal role for increased NO in the brain in behavioral modifications after stress exposure, the anxiogenic effect induced 24 h after an acute 2 h-restrain stress session, in view of modeling a trauma, was correlated with increased nNOS levels in the PL PFC. Moreover, the anxiogenic behavior was prevented by the administration of an nNOS inhibitor (N-propyl-L-Arginine; NPLA) in the same brain region [100]. Besides, the administration of L-arginine or L-NAME in the basolateral amygdala (BLA) prevented the long-term alterations in anxiety and depressive-like behavior induced by chronic stress [101]. Furthermore, adding to the complexity of NO and stress response, the administration of L-arginine (300 mg/kg) abolished the stress adaptive response in mice subjected to a 5-day low-intensity foot-shock stress protocol, while L-NAME (30 mg/kg) favored stress adaptation in mice submitted to a high-intensity foot-shock stress [102]. The nNOS overexpression, at least in the hippocampus of mice exposed to chronic mild stress, is crucial for the development of stress-induced depressive behavior [103], suggesting the involvement of nNOS in modulating coping behavior in response to stress. In line with this data, nNOS in the nucleus accumbens, a reward center, regulates susceptibility to social defeat stress and subsequent depression-like behaviors in mice [104].

Similarly, in the chronic stress model induced by corticosterone, there was an upregulation of nNOS mediated by ERK activation in the hippocampus. This upregulation was associated with reduced hippocampal glucocorticoid receptor (GR) levels, thereby altering the mineralocorticoid receptor (MR):GR ratio [105]. The impairment in the MR:GR ratio is directly involved in the hyperactivity of the HPA axis [109] but also in the susceptibility to developing mental disorders, including major depressive disorder (MDD) and PTSD, as a result of chronic stress or traumatic experiences [110,111].

The susceptibility to behavioral alterations induced by stress, as well as the potential regulation of nNOS, can at least partially be attributed to the modulation of NO-induced PTMs and their underlying pathways. Numerous studies have now demonstrated the essential role of S-nitrosylation in synaptic responses to physiological stimuli and under pathological conditions [112,113,114]. Therefore, this modification could facilitate cellular signaling in response to stress, offering the distinct benefit of enabling rapid alterations in the functionality of pre-existing proteins due to the numerous potential combinations of PTMs at various amino acid sites, which are also readily reversible. This versatility underscores the PTM’s potential pivotal role in adapting and fine-tuning cellular responses for survival and resilience in response to stress.

In this sense, nNOS could account for the altered sensitization of GR by PTMs in heat shock protein (Hsp)70/Hsp90 organizing protein (Hop)—also known as stress-induced phosphoprotein 1 (Stip-1)—a highly S-nitrosylated protein, abolishing its ATPase activity [115]. Since this complex acts by regulating the activation of GR, maintaining GR structure and ligand-binding affinity, S-nitrosylation may alter the ability of GR to provide both rapid and subtle responses to changing hormone levels while also maintaining GR in a nonaggregating, high-affinity state [116].

Furthermore, one of the main possible mechanisms underlying the effects of nNOS in stress might be the activation of DexRas1 through S-nitrosylation at Cys11 within the nNOS/NOS1AP/DexRas ternary complex [79,91]. DexRas-1 belongs to the Ras family of small monomeric G proteins, enriched in the brain, and was initially discovered in response to dexamethasone and stress [117]. Studies have shown that DexRas1 is essential for NMDAR-induced neurotoxicity mediated by NO, resulting in increased iron influx via the divalent metal transporter 1 (DMT1 or Slc11a2) [118,119]. Additionally, Dexras1-promoted intracellular iron homeostasis has been demonstrated to modulate excitatory currents and NMDAR function through the Src/protein kinase C (PKC) pathway, suggesting a role in the homeostatic control of neuronal excitability [120]. Interestingly, Src kinase is shown to be activated by S-nitrosylation at the residues Cys489, Cys498, and Cys500 mediated by NMDAR/nNOS responses [121]. Src has the major function of regulating glutamatergic neurotransmission through NMDAR upregulation by the phosphorylation of its NR2 subunits [122,123] and also modulates fear learning and synaptic plasticity in the amygdala [124]. Therefore, stress-induced overexpression of DexRas-1 mediated by nNOS and its upstream effects could directly contribute to the dysfunction in glutamate neurotransmission, a cardinal feature of stress-related psychiatric disorders, including PTSD [125].

Stress-induced activation of DexRas1 via nNOS is also accompanied by a reduction in the ERK–CREB signaling pathway in the brain, which is directly involved in fear memory acquisition, retention, and extinction learning. This pathway exerts important effects on synaptic plasticity in different brain regions, including the hippocampus and amygdala [126,127,128]. Although the mechanisms involving NO in response to stress and fear are not fully understood, it is reasonable to suggest that not only nNOS, but other molecules related to it, play an essential role in stress situations and the convergent pathological alteration of the glutamatergic system, ultimately influencing fear circuitry. Thus, additional research involving the NO system and activation of adjacent pathways through S-nitrosylation could enable a better understanding of the neurobiology involved in the formation and persistence of aversive memories.

### 3.1. The Interplay between Inflammatory Markers and NO in PTSD

One hallmark of a subset of PTSD patients is heightened cortisol sensitivity. Apart from this, PTSD is characterized by an enhanced immune response, including in the central nervous system (CNS) [129,130]. There is consistent evidence indicating the increase of several inflammatory markers in individuals who have experienced traumatic events, including C-reactive protein (CRP), IL-1β, IL-6, and tumor necrosis factor α (TNF-α) [131]. Levels of IL-6 in cerebrospinal fluid have also been found to be elevated in PTSD [132]. The inflammatory response in the absence of a pathogen or tissue damage, called sterile inflammation, is frequently linked to heightened exposure to stress [133]. These reactions are usually mediated by damage-associated molecular patterns (DAMPs) (Figure 3) [134], contributing to the local or systemic inflammation often seen in PTSD patients [135,136]. In addition, brain regions relevant to fear and anxiety, such as the amygdala, hippocampus, mPFC, ACC, and insula, are impaired by inflammation effects [137]. These inflammatory responses are mainly triggered by microglia cells, the resident immune cells of the brain, which are associated with (mal)adaptive responses to traumatic experiences [138,139]. Production of NO and other inflammatory mediators, such as TNF- α, by microglia are inhibited by cortisol [140]. Therefore, impaired cortisol signaling in PTSD could account for increased (neuro)inflammation, including high levels of NO.

One of the DAMPs released from microglia by stress exposure (sterile inflammation) is the high-mobility group box 1 (HMGB1) [141,142,143]. HMGB1 can act via activation of Toll-like receptor 4 (TLR4) or Receptor for Advanced Glycation End Product (RAGE) in microglia, resulting in the release of several inflammatory mediators [144]. In the brain, its levels can be increased by severe stress, contributing to neuroinflammation [145,146]. Patients exposed to trauma who then developed PTSD presented high levels of plasma HMGB1 when compared to those who did not develop the disorder [147], indicating that this DAMP could play a role in PTSD development. HMGB1 is shown to be S-nitrosylated at Cys106. Interestingly, S-nitrosylated HMGB1 secretion induced more profound microglial activation and neuronal death than unmodified HMGB1 [148]. One of the main regulators of HMGB1 expression, modification, and release is the JAK-regulated activation of signal transducer and activator of transcription (STAT)1 and STAT3 [149]. STAT3 S-nitrosylation at Cys259 is shown to regulate microglial proliferation and inhibition of STAT3 (Tyr705) phosphorylation [150]. Considering Arginase-1, a marker often expressed by the anti-inflammatory phenotype of microglia induced by IL-4, directly competes with NOS synthases for the substrate L-arginine [151], microglial polarization is also linked with NO metabolism.

P2X7 receptor (P2X7R) is an ATP-gated nonselective cationic channel, majorly involved in the polarization of microglial cells, production of HMGB1, and activation of IL-1β maturation/release mediated by NLRP3 inflammasome assembly and caspase-1 activation (Figure 3) [152,153]. It has been shown that central delivery of a P2X7 inhibitor (A-438079) administered immediately after single prolonged stress (SPS), a model of PTSD, and repeated for 7 days prevented stress-induced extinction learning impairment and neuroinflammation in the ventral hippocampus [154]. P2X7R is composed of a large extracellular domain, two transmembrane domains, and cytoplasmic N- and C-termini. An extracellular domain has ten cysteine residues forming five intrasubunit disulfide bonds (SS1–SS5) [155], which are needed for the trafficking of P2X7R to the cell surface [156] and the recognition of surface epitopes [157]. Recently it has been shown that P2X7 is S-nitrosylated, partially by the transfer of NO by protein disulfide isomerase enzyme. This modification facilitates the disulfide bond formation of P2X7 and thereby maturation, increasing its trafficking to the cell membrane [156,158]. P2X7 and P2Y1, but not P2X1, are co-immunoprecipitated with nNOS in the mPFC, hippocampus and striatum. In addition, the treatment with a non-selective P2X inhibitor systemically inhibited the activation of nNOS in the mPFC e hippocampus of mice [159].

While iNOS activation is directly linked with inflammatory responses, nNOS also seems to play a crucial role in regulating this process in the brain, especially when triggered by NMDAR excitotoxicity [160]. Inflammation regulated by nNOS is believed to occur through the activation of cyclooxygenase-2 (COX-2), an initiator of inflammatory responses that converts arachidonic acid into pro-inflammatory prostaglandins (PGs), resulting in the production of chemokines and cytokines (Figure 3) [161]. Evidence indicates that NMDAR-induced excitotoxicity relies on the S-nitrosylation of COX-2 and its subsequent activation, a process facilitated by the interaction with the PDZ domain of nNOS. This mechanism differs from the conventional activation of COX-2 by iNOS, which occurs through the enzyme’s active site, suggesting a dual effect of COX-2 activation due to NO production [162]. Considering that the PDZ domain is unique to nNOS among the three NO synthases, this activation mode underscores effective and selective nNOS-induced neuroinflammatory conditions by the overactivation of NMDAR (Figure 3).

Interestingly, as reported by Wang et al. (2018), exposure to SPS revealed higher *COX-2* mRNA and protein expression accompanied by higher TNF-α, IL-6, PGE2 and NO levels in the hippocampus; meanwhile, treatment with a selective COX-2 inhibitor, celecoxib, attenuated the consequences of stress exposure in rats [163]. Similarly, administration of different selective COX-2 inhibitors (LM-4131, lumiracoxib-LMX, or celecoxib) 6 h after the acquisition of fear memory reduced the expression of contextually conditioned fear in mice. In the same study, LM-4131 significantly decreased the intrinsic excitability of BLA neurons induced by foot-shock stress [164], corroborating the relationship between PGE_2_ and glutamatergic synaptic transmission. S-nitrosylation is a physiological regulator of the transcription factor NF-κB [165]. Among several immune and inflammatory responses mediated by NF-κB, there is upregulation of all major NO synthases. It is shown that NO-induced S-nitrosylation of Cys118 on p21Ras is responsible for the initiation of this signal transduction [166,167]. Additionally, S-nitrosylation of SIRT1 is likely to have a critical role in the inflammatory cascade, comprising activation of iNOS/nNOS and subsequently activation of the p65 subunit of NF-κB, and p53 [168]. However, iNOS-induced S-nitrosylation of p65 at Cys62 inhibits NF-κB-dependent gene transcription and is also a conserved residue in other Rel-homology domain-containing proteins, including p65, p52, p100, p105, and c-Rel [169]. This evidence suggests that S-nitrosylation could account for modulating the phosphorylation-dependent activation of NF-κB [170].

Regarding the relationship between stress and NF-κB, chronic stress or corticosterone induce overactivation of hippocampal NF-κB, which promote anxiogenic behavior, potentially mediated by the increased association of nNOS-NOS1AP-Dexras1 complex [171]. NF-κB inhibition after memory reactivation impairs the retention of amygdala-dependent auditory fear conditioning (AFC) in the BLA. This effect can be reversed by pre-treatment with sodium butyrate, which can act as a non-selective HDAC inhibitor. These findings imply that NF-κB activity in the BLA is required for memory reconsolidation, at least in the AFC [172].

IL-1β and other cytokines, such as IL-6, are increased in the brain in animal models of PTSD [173,174,175]. Moreover, IL-1β activates p38 MAPK in astrocytes and hippocampal neurons [176], a pathway linked to synaptic plasticity that contributes to hippocampal NMDAR-dependent long-term depression (LTD), thereby influencing long-term memory in vivo [177]. Notably, IL-6 possess the ability to modulate fear learning and the processing of stress-related experiences. IL-6 release in response to fear memory is essential for maintaining fear memories following retrieval in mice [178]. Moreover, individuals with PTSD had lower NO synthesis capacity associated with higher IL-6, TNF-α, and PTSD symptom severity, and higher CRP levels [179]. Ultimately, IL-6 levels have the potential to dysregulate various signaling pathways in brain regions associated with PTSD [180], making it a significant contributing factor to the development of its symptoms.

Interestingly, inflammation can disrupt several neurotransmitter systems, including glutamate and GABA, and individuals with PTSD presented elevated levels of glutamate, which contribute to symptoms like anhedonia. Therefore, there seems to exist a link between increased levels of immune mediators and glutamate observed in PTSD patients, modifications that could contribute to the disruption of fear mechanisms.

Heightened expression of iNOS and phospho-p38 in the hippocampus are observed in the enhanced SPS model of PTSD [175]. However, contrary to these findings, extinction of inhibitory avoidance (IA) long-term memory requires activity of p38 in the CA1 region of the dorsal hippocampus. Inhibition of p38 blocked memory reacquisition after extinction [181], and blocked short- and long-term memory formation [182]. This implies a contrasting role of p38 in fear memory, which could be dependent on the kind of memory evaluated, but also could be due to different pathways activated downstream. Although p38 plays a role in inflammation and memory formation, it also participates in cell death, which will be further discussed.

Another source of inflammatory mediators after stressful events is the interference with cellular metabolism and production of reactive species, which will be discussed further in the next topic. For instance, the release of IL-1β can be triggered by the production of mitochondrial reactive oxygen species (mtROS) [16]. Moreover, time-dependent sensitization (TDS), a model of PTSD, led to increased hippocampal NOS expression, especially iNOS, along with downregulation of hippocampal NMDAR and dysregulation of inhibitory GABA pathways [105], suggesting major roles for inflammation and nitric oxide interplay in PTSD behavioral outcomes.

Taken together, these pieces of evidence underscore the intricate interplay of NO signaling with inflammatory pathways, oxidative stress and modulation of neurotransmitters, suggesting potential targets for pharmacotherapy in the context of PTSD.

### 3.2. Cellular Metabolism and Nitrosative Stress

Production of mtROS, which is related to increased cytokine production, can substantiate the framework of inflammatory responses under chronic exposure to trauma [133,183]. ROS are typically associated with oxidative stress that generates O2^−^, OH, and H_2_O_2_ [184]. Nitrosative stress denotes an imbalance between the production and neutralization rate (i.e., detoxifying antioxidant components) of reactive nitrogen species (RNS) (discussed in more detail below) and ROS within the biological system. This imbalance can affect the molecular components of the cell, including lipids, proteins/enzymes, carbohydrates, and nucleic acids, resulting in detrimental effects that further propagate alterations in cell signaling, gene expression, and apoptosis (Figure 3) [185]. The brain is well-known for its vulnerability to nitrosative stress due to factors such as high oxygen consumption, its lipid-rich structure, and the presence of redox-active metals like copper and iron, along with limited antioxidant defenses [186]. Consequently, nitrosative stress can result in adverse effects including increased blood-brain barrier permeability, impaired neurogenesis and synaptic plasticity, alterations in neurotransmission, changes in neural morphology, and disruptions in overall cellular signaling [185]. These factors collectively contribute to the pathophysiology of PTSD, although clinical evidence remains contradictory considering the challenges of measuring nitrosative components and biological markers [187,188]. Interestingly, scavenging ROS with N-acetyl cysteine, a potent antioxidant, in LA before extinction training attenuates fear renewal [75].

NO, a highly reactive and unstable molecule, is involved in many different processes in the CNS. Therefore, a delicate balance exists between its physiological actions and cytotoxic effects. At lower concentrations, its physiological functions play a pivotal role in cellular metabolism, whereas at higher levels, it engages with free radicals, including superoxide (O^2−^) and hydrogen peroxide (H_2_O_2_), thereby generating RNS. RNS are derived from one common precursor, NO, and encompass nitrite (NO^2−^), S-nitrosothiols, dinitrosyl-iron complexes, and peroxynitrite (OONO^−^) [189]. NO^2−^ and nitrate (NO^3−^) [190] are the products of NO oxidation. Furthermore, peroxynitrite is synthesized by the reaction between NO and superoxide; considering its potent oxidant profile and the absence of effective detoxifying agents, it can affect a wide range of cellular cascades. It can react with carbon dioxide (CO_2_), propelling the formation of the unstable nitrosoperoxycarbonate anion (ONOOCO_2_) that directly reacts with proteins, promoting its carbonylation [191]. Peroxynitrite affects low molecular weight proteins [192], metal centers, as well as protein-bound thiols, promoting alterations in the conformation and activity of proteins [193].

As previously mentioned, stress induces glutamate excitotoxicity, and dysfunction in glutamate neurotransmission is a major signature of PTSD. Glutamate excitotoxicity is a deleterious event which might play a significant role in neuron injury and thereby, in neurotransmission pathways, which contributes to re-experiencing symptoms and other memory abnormalities [125,194]. Interestingly, the resulting nitrosative imbalance of excitotoxicity can favor other PTMs, such as tyrosine nitration, which was demonstrated to be an important regulatory mechanism of NMDAR [195]. Thereby, peroxynitrite-induced nitration can regulate NMDAR by increasing affinity of the ion-channel and glutamate sites, and by exposing additional ion-channel sites. This modification potentiates the Ca^2+^ influx contributing to enhanced excitotoxicity, therefore promoting a vicious loop [196]. NO also engages the PTM glutathionylation, catalysing the addition of GSH, or other low-molecular-weight thiols, to the cysteine sulfhydryl residues of proteins [191]. In the mitochondria, the transfer of the nitrosyl group from the heme iron of cytochrome-c to GSH produces GSNO (Figure 3). GSNO stands out as the most prevalent S-nitrosothiol and serves as the primary endogenous source of NO for proteins throughout the cell [197]. Once GSNO is generated, it undergoes translocation to various subcellular locations and engages in transnitrosylation, affecting interacting proteins such as NF-κB, STAT3, protein kinase B (PKB), epidermal growth factor receptor (EGFR), and insulin-like growth factor 1 receptor (IGF-1R) [198,199]. Relevant studies involving individuals with PTSD have revealed a 23% increase in GSH concentrations in two specific brain regions, the dorsolateral (DL) PFC and ACC, detected by single-voxel proton magnetic resonance spectroscopy (MRS), suggesting elevated production of RNS and ROS [200]. Similar levels were also observed in the ACC region of older adults with a lifetime history of depression [201].

A frequently observed reaction prompted by RNS is lipid peroxidation, a process in which oxidants target lipids containing carbon-carbon double bonds, leading to oxygen insertion and the formation of lipid peroxyl radicals and hydroperoxides [202]. Malondialdehyde (MDA) represents both a by-product of lipid peroxidation and an indicator of damage to the cell membrane. Elevated levels of MDA function as a biomarker reflecting the efficiency of the antioxidant defense system and the extent of oxidative damage; both lipid peroxidation and oxidative stress can be assessed by measuring the concentration levels of MDA. Increased levels of lipid peroxidation are associated with depression and chronic stress disorders, such as PTSD [203]. Furthermore, a study conducted with a genetic model of generalized anxiety, Carioca high-conditioned freezing (CHF) rats, showed that free radical concentrations and MDA levels were significantly higher in the cortex, cerebellum, and preeminently in the hippocampus of these rats compared with Carioca low-conditioned freezing (CLF) rats. These findings strongly suggest the involvement of the redox system in fear conditioning and highlight the hippocampus as one of the key brain structures implicated in this oxidative stress imbalance [204].

RNS are also capable of inducing depurination of DNA. Specifically, peroxynitrite can react with guanine, producing 8-nitroguanine (8-NG) [205]. Guanine, due to its minimal oxidation potential, is particularly susceptible to attack by various reactive species. The highly redox-active nucleic acid intermediates, such as the deprotonated guanine neutral radical (G-H), generated by nitrosative stress, have the potential to cause DNA mutations and facilitate the formation of inter- and intrastrand DNA crosslinks. Such a process serves as a crucial indicator of RNS-mediated DNA oxidation, which ultimately results in DNA fragmentation, giving rise to aberrant biological activities [206]. Interestingly, the intracerebroventricular administration of 8-nitro-cGMP resulted in cognitive deficits, represented by a decrease in context-dependent fear memory in mice. This effect was accompanied by an increase in the S-guanylation of proteins, including synaptosomal-associated protein (SNAP-25), which were correlated to the altered formation of synaptic SNARE complexes which play a key role in synaptogenesis and exocytosis of neurotransmitters [207]; and decrease in complexes containing complexin, which bind to SNARE complexes and also act as regulators of synaptic vesicle fusion and neurotransmitter release [207], in the hippocampus. In summary, these findings strongly indicate that the accumulation of 8-nitro-cGMP in the hippocampus can impact neurotransmission, including aversive memory acquisition [208].

### 3.3. Role of NO in Cell Death and Excitotoxicity

An impaired physiological microenvironment, characterized by inflammatory and oxidative markers, decreases the ability of cells to survive. This condition leads to reduced neuroplasticity and increased cell death, mainly through necrosis (dichotomously referred to as apoptosis) [209]. Regarding PTSD, increased apoptosis has been observed in key brain regions, including the hippocampus, amygdala, and mPFC of animals subjected to a PTSD model [210]. NO plays a pivotal role in this process by activating members of the mitogen-activated protein kinase (MAPK) pathway, including c-Jun *N*-terminal kinase (JNK), p38-kinase, and ERK1/2. Specifically, the activation of p38 through NMDAR involves the recruitment of NOS1AP to nNOS. NOS1AP interacts with MAP kinase kinase 3 (MKK3), which is essential for excitotoxic activation of p38 [211]. Under stress conditions, the activation of JNK and p38 triggers a pro-apoptotic NO signal, leading to the phosphorylation of Ser15 of p53. Subsequently, p53 accumulates, serving as an early indicator of NO-induced apoptosis and contributing to caspase activation (Figure 3) [212,213,214]. In contrast, ERK activity serves as an anti-apoptotic signal and is associated with cell survival [213].

Caspase family proteins, the primary regulators of apoptosis, exist as inactive zymogens in resting cells. One of the upstream signaling pathways leading to caspase activation involves the release of cytochrome C and other apoptogenic factors from damaged mitochondria (Figure 3) [215]. There is increasing evidence that the pro-apoptotic effect of NO may result from the inhibition of mitochondrial respiration, which leads to membrane potential reduction, transition pore opening, and release of cytochrome C. Caspases can propagate apoptotic signaling by cleaving/activating other caspases, such as caspase-8, -9, and -10. Alternatively, other caspases, including caspase-3, -6, and -7 can execute the terminal events in apoptosis by cleaving specific target proteins, resulting in protein degradation and DNA fragmentation [216]. Exposure to the PTSD model SPS induced apoptosis in hippocampal neurons of rodents, accompanied by increase of caspase-9 and caspase-3 activities [217]. This is in line with the observed dysfunction and low volume of the hippocampus in PTSD patients (for review, see [218]).

Following exposure to stress, a wide range of apoptotic stimuli can augment NO production, either through the induction of iNOS or activation of nNOS. Most caspases are susceptible to redox modification due to the presence of a catalytically reactive cysteine moiety that can be S-nitrosylated in the presence of NO. Activation of cell death pathways induces both cleavage and selective denitrosylation of the catalytic cysteine residue of caspase-3 [219]. Caspase-3 can transnitrosylate its binding partner, X-linked inhibitor of apoptosis (XIAP), at Cys450 in the RING domain. This process suppresses XIAP’s E3 ligase activity, consequently diminishing its antiapoptotic function in neuronal cells. In neurons exposed to pathologically high levels of NMDA, there was an accumulation of S-nitrosylated XIAP (SNO-XIAP) (Figure 3) [220]. Furthermore, transnitrosylation from procaspase-9 to XIAP at Cys325, mediated by the Trx1 system, has been observed to consequently cause apoptosis [221]. Therefore, transnitrosylation of XIAP by SNO-caspase helps explain why nitrosative stress is often accompanied by cellular death [222] although physiological levels of NO might promote cell survival [209].

NO produced in response to cellular stress also triggers glyceraldehyde-3-phosphate dehydrogenase (GAPDH) S-nitrosylation at Cys150, a critical site for the enzyme’s catalytic function [223]. This modification abolishes the catalytic activity of GAPDH but enables it to interact with Siah1, also a member of the RING-finger-containing E3-ubiquitin-ligase family, which possesses a nuclear localization signal (NLS). Treatment of HEK293 cells with NO donors enhances GAPDH binding to Siah1 through Cys150 [224]. Subsequently, Siah1 escorts GAPDH to the nucleus, eliciting cell death (Figure 3) [225]. Siah is widely expressed in the brain [226], with high expression of its mRNA observed in pyramidal neurons of the hippocampus and Purkinje cells of the cerebellum [227]. In addition to Siah1, GAPDH interacts with GAPDH’s competitor of Siah protein enhances life (GOSPEL), a cytoplasmic protein that interferes with GAPDH’s interaction with Siah1, thereby preventing apoptosis. It has been observed that S-nitrosylation of GOSPEL at Cys47 significantly enhances its binding to GAPDH, thus preventing the translocation of GAPDH into the nucleus (Figure 3). This evidence suggests that GOSPEL may have a neuroprotective effect, as its overexpression prevents NMDAR-glutamate excitotoxicity [228]. The competition between GOSPEL and Siah1 for GAPDH binding may represent a regulatory mechanism that maintains cellular homeostasis in response to stressors that may be disrupted in neurodegenerative diseases and stress-related disorders, such as PTSD.

While GAPDH has traditionally been recognized as a housekeeping glycolytic enzyme, recent research has revealed its multifunctional nature [229,230]. Over the years, reports have emerged suggesting that GAPDH exhibits other activities. These include being associated with cell death induced by oxidative stress [231] and functioning as a uracil-DNA *N*-glycosylase (UNG) involved in DNA repair [232]. UNG KO mice subjected to a folate-deficient diet exhibit degeneration in the CA3 pyramidal neurons, accompanied by reduced levels of BDNF and GSH. These animals also display cognitive impairment, as well as anxiety- and despair-like behavior [222]. Additionally, GAPDH has been shown to bind to the inositol 1,4,5-triphosphate receptor (IP3R), thereby modulating Ca^2+^ release through NADH production [233]. IP3R dysregulation may contribute to memory deficits through its effects on Ca^2+^ signaling, hippocampal function, and synaptic plasticity. In a study investigating the role of IP3R in contextual fear memory consolidation, retrieval, reconsolidation, and extinction; pharmacological inhibition of IP3R in the hippocampus impaired contextual fear memory consolidation and retrieval [234].

Furthermore, as previously mentioned, it is widely recognized that the activation of NMDAR by glutamate can lead to neuronal cell death through nNOS activation and subsequent overproduction of NO [235]. This neurotoxicity was decreased in neuronal cultures upon the inhibition of nNOS [236]. In cerebellar granule neurons, cultures from WT, but not from nNOS KO mice, displayed increased interaction between GAPDH and Siah1 following glutamate exposure [237]. Moreover, there is evidence that NMDAR-elicited neuronal death can be prevented by depleting Siah1 or GAPDH using RNA interference (RNAi) techniques [224], suggesting that NO/GAPDH/Siah1 signaling plays a role in excitotoxic neuronal cell death. Hence, stimuli that induce cellular death by disrupting microenvironment homeostasis might contribute directly to reducing neuroplasticity and altering neurotransmission pathways. These processes are essential for extinction learning, and, indirectly, for promoting the persistence of fear memories.

### 3.4. nNOS and the Regulation of Synaptic Transmission and Plasticity

Changes in functional connectivity and synaptic transmission have been suggested to be an important feature of PTSD pathophysiology. For example, reduced prefrontal glutamatergic synaptic strength [238] and higher cortical metabotropic glutamate receptor 5 (mGluR5) have been described in PTSD patients [239]. These patients also show higher levels of glutamate and lower levels of the integrity neuronal marker N-acetyl aspartate (NAA) [194]. Regarding genetic predisposition, variation in a glutamate transporter gene (SLC1A1), which codes for the excitatory amino acid transporter 3 (EAAT3) and excitatory amino acid carrier 1 (EAAC1), has been associated with risk for developing PTSD and greater symptom severity [240]. These pieces of evidence support the proposal that glutamate synaptic changes and altered glutamate availability are crucial events in trauma and stress-related psychopathologies.

Despite the close interplay between nNOS and NMDAR, growing evidence suggests their role in regulating signal transmission through the release of neurotransmitters in presynaptic neurons and the modulation of postsynaptic receptors. In this sense, it was initially observed that NO can promote the synchronous release of synaptic vesicles by increasing the formation of SNARE protein complexes (soluble N-ethylmaleimide-sensitive factor attachment protein receptors), which includes vesicle-associated membrane protein 2 (VAMP2, synaptobrevin2), SNAP-25, and syntaxin 1a, while also inhibiting n-sec1 to syntaxin 1a [241]. Later on, it was demonstrated that S-nitrosylation of Cys145 on the cytoplasmatic domain of syntaxin 1a selectively disrupt the classical binding mode of Munc-18, which facilitates the syntaxin 1 engagement with SNARE complex and neurotransmitter exocytosis [242].

Under resting conditions, synaptic vesicles are stored in the cytoplasm of neurons, with some attached to the active zones on the presynaptic plasma membrane. The regulation of these vesicles is mediated by SNARE proteins (Figure 4), which form a tight complex with their α-helical SNARE motifs. The energy released during the zippering of these α-helical SNARE motifs into the SNARE complex allows the fusion of vesicle and plasma membranes. Thus, vesicles of neurotransmitters mediated by the SNARE complex reside in a low-energy configuration within the plasma membrane under the control of the active zone assisted by Sec1/Munc-18 proteins (docking step). Following an action potential and influx of Ca^2+^, the synchronous Ca^2+^ sensor, synaptotagmin 1, triggers the fusion of the synaptic vesicle membrane with the active zone membrane (priming step). This fusion leads to pore opening and the subsequent release of neurotransmitters into the synaptic cleft [243,244].

The speed and effectiveness of neurotransmission also depend on vesicle recycling and the traffic of vesicles between different pools (readily releasable, reserve, and resting). After fusion, the N-ethylmaleimide-sensitive factor (NSF), an ATPase, and SNAPs disassemble the SNARE complex to recharge individual SNARE proteins for further release cycles. It has been shown that NSF is susceptible to nitrosylation at the residues Cys91 and Cys264, promoting its inactivation possibly through the prevention of NSF ATPase activation by SNAPs (Figure 4) [245].

Altogether, this evidence suggests that NO may play a regulatory role in the exocytosis of neurotransmitters, facilitating neurotransmitter release but also reducing the speed of SNARE reassembly in a reversible manner. Interestingly, long-term fear memory formation involves alterations in synaptic efficacy mediated by chemical and/or structural changes in synaptic transmission, inducing neuronal hyperactivity, that might be affected by neurotransmitter exocytosis [246]. Recently, it was suggested that AFC facilitates neurotransmitter release at LA to BLA synapses, showing shared mechanisms with conditioning and electrically induced LTP [247]. Simultaneously, stress also reduces synaptic inhibition. These changes in synaptic function enable amygdala neurons to remain in a hyper-responsive state during subsequent emotional experiences, potentiating the formation of stronger fear memories [248]. This evidence could suggest a way in which NO promotes the consolidation and persistence of fear memories.

NSF, along with thorase, also influences excitatory synaptic transmission by regulating the internalization and insertion of AMPARs into postsynaptic membranes. Thus, AMPAR trafficking elicited by NMDAR neurotransmission is mediated by a cascade involving NMDAR activation of nNOS, leading to the nitrosylation of thorase at Cys137. Thorase acts by disassembling GluA2/glutamate receptor-interacting protein 1 (GRIP1) complex in an ATP-dependent manner. When thorase is S-nitrosylated, its ATPase activity is inhibited, resulting in the stabilization of the GluA2/GRIP1 complex and enhancing its endocytosis (Figure 4). S-nitrosylation of thorase also augments its interaction with NSF, promoting NSF transnitrosylation and subsequent activation [249]. Activated NSF can bind to GluA2/protein interacting with the C kinase 1 (PICK1) complex, dissociating PICK1 from GluA2, thereby enhancing NSF-mediated insertion of GluA2 at the synapse [121].

Corroborating the idea that NMDAR-induced nitrosylation regulates AMPAR surface expression, it has been demonstrated that stargazin is also a target of this phenomenon. Stargazin, a member of the transmembrane AMPAR regulatory protein (TARP) family, acts as an auxiliary subunit of all AMPAR subtypes [250]. Under basal conditions, nitrosylation of stargazin at Cys302 by NO increases its binding to GluR1, thereby enhancing surface expression of the receptor [251]. PSD95 also regulates AMPAR through its interaction with stargazin, corroborating the recruitment of AMPAR to the synapse [252]. Additionally, the GluA1 subunit, besides being S-nitrosylated under basal conditions, undergoes increased S-nitrosylation upon NMDAR stimulation. Therefore, NMDAR-induced S-nitrosylaton of Cys875 facilitates the phosphorylation of Ser831, increasing the conductance of this channel. Furthermore, S-nitrosylation of Cys875 also seems to facilitate the endocytosis of the receptor by enhancing its binding to the clathrin adaptor AP2, which, in turn, regulates its surface expression, a critical step in the modulation of LTD [251].

As previously mentioned, PSD95, the main component of postsynaptic densities, binds nNOS, facilitating the linkage of NMDAR-mediated neurotransmission to activation of nNOS. It has been shown that PSD95 is physiologically S-nitrosylated at Cys3 and Cys5 in a reciprocal relationship with palmitoylation, indicating that NO normally impacts major functions of PSD95 [56]. Glutamate-NMDAR neurotransmission triggers the depalmitoylation of PSD95, as documented by El-Husseini et al. (2002) [253]. S-nitrosylated PSD95 competes with palmitoylation, effectively blocking free cysteines and maintaining PSD95 in a depalmitoylated state. Interestingly, it was shown that atypical crosstalk between S-palmitoylation and S-nitrosylation of proteins involved in synaptic transmission might be one of the main events associated with stress, leading to synaptic network destabilization [57]. Consequently, an orchestrated balance between nitrosylation and other PTMs might play a role in cellular signal transduction and could be affected by stressful events, thereby impacting synaptic transmission.

Altogether, the presented data indicate that NO might mediate excitatory synaptic transmission in the brain, through the regulation of NMDAR/AMPAR interplay. This fast and highly dynamic process of AMPA shuttling in and out of synapses in an activity-dependent manner fundaments LTP and LTD processes and consequently synaptic plasticity, which modifies synaptic strength and supports cellular forms of learning. Studies indicate that nNOS mediates both early and late stages of fear memory consolidation through sequentially orchestrated changes in synaptic plasticity facilitated by AMPAR and NMDAR in the BLA [254]. These changes in synaptic AMPAR density induced by NMDAR stimulation might be affected by stress and provide a key mechanism for activity-dependent modulation of synaptic strength. This phenomenon also underlies different aspects of fear memory. For instance, in the amygdala, AMPARs are incorporated into postsynaptic neurons during fear acquisition [255,256]. Meanwhile, AMPAR removal at excitatory synapses underlies the extinction process [257,258]. Finally, in the hippocampus, AMPAR delivery is required for fear consolidation, while blocking its endocytosis impairs extinction learning [259], suggesting that AMPARs might also be required for fear extinction [260,261].

Neurotrophic factors may also modify glutamate signaling by changing the expression of NMDAR/AMPAR and Ca^2+^-regulating proteins [262]. Neurotrophins, including BDNF, NGF, neurotrophin-3 (NT-3), and NT-4, belong to a closely related family of peptides. Collectively, they play a crucial role in differentiation, cell survival, and synapse formation within the CNS [263]. Rattiner et al. (2004) have shown that BDNF, among other neurotrophins, is specifically upregulated in BLA after fear acquisition. BDNF modulates the induction and maintenance of LTP through binding to its receptor TrkB [264]. Additionally, both BDNF and TrkB, along with epigenetic regulation of the *BDNF* gene during fear memory consolidation, were increased in the SPS model compared to naive rats. This suggests that BDNF might be associated with stress-induced long-lasting fear memory [265].

When BDNF binds to TrkB, it induces a conformational change in the TrkB dimer, leading to the autophosphorylation of multiple tyrosine residues within TrkB, including Tyrosine-816 (Tyr816). This autophosphorylation, in turn, triggers the recruitment of adaptor molecules to the phosphorylated tyrosine residues, subsequently activating several intracellular pathways. These pathways include the Ras-MAPK pathway, the phosphatidylinositol 3-kinase (PI3K)/PKB pathway, and the phospholipase C γ (PLC-γ) pathway [266]. Furthermore, nitration of TrkB impairs the PLCγ cascade by preventing Tyr816 phosphorylation and by reducing PLCγ binding to the nitrated TrkB, ultimately leading to clathrin-dependent endocytosis of TrkB and subsequent translocation of this receptor towards lysosomal degradation [267], potentially reducing synaptic plasticity phenomena.

The regulation of fear learning and amygdala synaptic plasticity depends on downstream activation pathways. For instance, the acquisition of conditioned fear responses to both context and auditory cues relies on the PLC-γ pathway, while the shc (Src homology 2 domain-containing-transforming protein C1) path plays a crucial role in the consolidation of conditioned fear response [268]. In contrast, administration of BDNF into the IL, but not the PL mPFC can induce extinction of older and recent fear memories [269]. Therefore, this evidence suggests a dual role for BDNF and related pathways in the processing of fear memories and PTSD, with its effects mainly dependent on the time course and the specific brain regions involved.

Several studies have shown that BDNF upregulates nNOS expression or NO production in neural progenitor cells, astrocytes, and neocortical neurons, suggesting that BDNF induces the production of NO and subsequent S-nitrosylation of Cys262 and Cys274 on HDAC2 in embryonic cortical neurons. The S-nitrosylated HDAC2, in turn, promotes chromatin remodeling, dendritic growth and branching, and activation of genes associated with developmental processes [270]. Interestingly, the interplay between NO and BDNF appears to be neuron type-dependent within the CNS, since exogenous BDNF treatment induces NO production in neocortical neurons but not in hippocampal neurons [271]. Moreover, NO seems to negatively modulate BDNF function. For instance, inhibiting NOS using L-NAME induces BDNF release by cultured hippocampal neurons and an increase in hippocampal BDNF mRNA in vivo which could indicate a negative feedback loop.

Compelling evidence suggests that NO, primarily produced by nNOS, leads to the downregulation of CREB transcriptional activity [272]. This downregulation, which exhibits an anti-proliferative effect, has been observed in both the hippocampal dentate gyrus and the subventricular zone [273,274]. It is interesting that PTSD patients showed reduced hippocampal volume relative to control subjects [275]. Typically, CREB is activated through Ser133 phosphorylation in response to external signals, such as those from neurotrophins or neurotransmitters [276]. Interestingly, CREB activation by BDNF appears to be modulated by an NO-dependent signaling pathway involving S-nitrosylation of components within the CREB DNA-binding complex [277]. In this context, although the NO/GAPDH/Siah1 signaling pathway plays a role in cell death, it also promotes CREB-mediated dendrite outgrowth through epigenetic chromatin-remodeling regulatory mechanisms [278], indicating the broad effects mediated by NO.

It has been demonstrated that stress-induced activation of the BLA via glutamatergic projections to the dorsal hippocampus is directly involved in up-regulation and phosphorylation of GRs, increased HDAC2 expression, and reduced expression of memory-related genes in the hippocampus. These effects are mediated by cyclin-dependent kinase 5 (CDK5) activation [279]. Interestingly, CDK5 can be activated by S-nitrosylation and can form a complex with nNOS. CDK5 also modulates nNOS activity by phosphorylating serine residues in nNOS [280]. Among its various targets, in excitotoxicity conditions CDK5 activation induces the phosphorylation of NMDAR, which is a primary intracellular event underlying hippocampal CA1 cell death [214]. The coactivator of CDK5, p35, has been shown to bind PSD95, forming a complex with the NMDAR and nNOS, dynamically regulating its clustering in the synapse [281]. Thus, nNOS also promotes the S-nitrosylation of p35 leading to its ubiquitin/proteasome-dependent degradation, which also can indirectly impact CDK5 activity [282]. In this sense, using p35 conditional KO mice (p35cKO) in hippocampal pyramidal neurons, showed that CDK5/p35 in excitatory neurons is important for hippocampal synaptic plasticity and associative aversive memory retention [283]. Therefore, the regulation of CDK5 by nNOS may be directly implicated in the consequences of stress in the BLA-hippocampus pathway, contributing to the persistence of aversive memory. Additionally, the cascades of cell death and the reduced expression of plasticity-related genes might be related to the difficulty in extinguishing fear and the acquisition of a new memory, reinforcing the dual role of synaptic plasticity in PTSD.

## 4. Conclusions

NO has been extensively associated with fear-related mechanisms, particularly through the activity of nNOS. However, its regulatory influence extends beyond simple NOS activation and NO production, encompassing various cellular mechanisms and pathways. In this sense, this review explored the intrinsic role of NO and its many associated pathways, especially in conjunction with nNOS, in the regulation of fear responses, stress, and their implications in pathophysiological alterations seen in PTSD. By unraveling the complexities of nNOS signaling, this study opens new avenues for understanding the profound impact of NO and NO-induced PTMs on the mechanisms affecting fear memory circuitry, offering potential therapeutic insights into fear-related disorders. These mechanisms, ranging from stress response and neuroinflammation to oxidative stress, cell death, and synaptic transmission/plasticity, constitute a crucial aspect of the physiological underpinnings of PTSD and related conditions.

Although, to our knowledge, no drug directly targeting NOS isoforms has been investigated in clinical trials for fear or anxiety-related disorders, one NOS inhibitor (Ronopterin-VAS203) and one Discs large homolog 4 inhibitor/NMDAR antagonist (Nerinetide-NA1 tat-NR2B9c) are currently being tested in Phase 3 clinical trials for traumatic brain injury (TBI) and/or ischemia [284]. If their efficacy and safety are confirmed in these trials, it could open the possibility of testing them in other conditions, including PTSD.

## Figures and Tables

**Figure 1 molecules-29-00089-f001:**
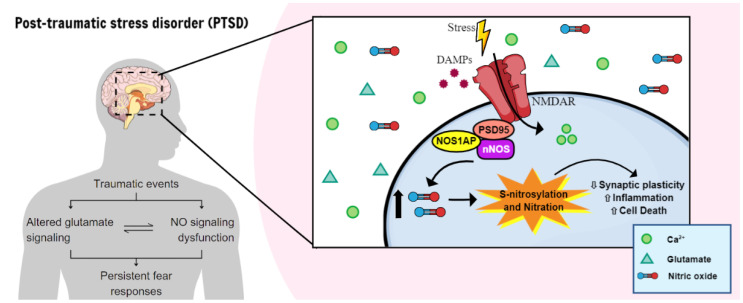
The involvement of NO signaling in PTSD pathophysiology. Persistent fear responses and altered glutamate signaling, essential for synaptic plasticity and memory formation, are key features of PTSD, resulting from traumatic experiences. Activation of the NMDAR can trigger the formation of a complex comprising PSD95, nNOS, and NOS1AP, which is pivotal for the activation of nNOS and NO production, that, in turn, induce posttranslational modifications (PTMs), such as S-nitrosylation and nitration. Together, they alter protein function and structure for intracellular signaling, activating downstream pathways that modulate neuronal signaling, including synaptic plasticity/transmission, inflammation, and cell death.

**Figure 2 molecules-29-00089-f002:**
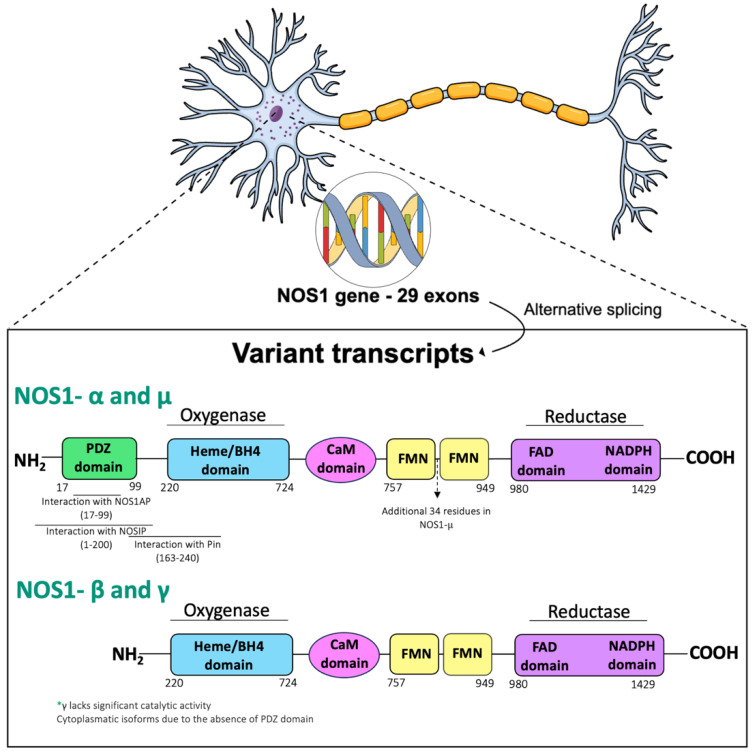
Major products of alternatively spliced nNOS mRNAs: nNOS-α, β, μ and γ. The *NOS1* gene comprises 29 exons, 28 introns, and these DNA elements are interspersed over a 250 Kb genome. All but exon 1 are translated to generate the most abundant isoform in the brain, nNOSα, a 150-kDa protein containing a PSD-95/discs large/ZO-1 homology domain (PDZ) that anchors this isoform to neuronal membranes through interactions with PSD95 and NMDA receptor. nNOSμ has a similar structure, but this variant contains a unique 102-base pair (34 amino acids) insert between the CaM and FMN binding domains and is majorly expressed in the heart and smooth muscle. nNOSβ translation is initiated at exon 1a generating a 136-kDa protein; meanwhile, the translation of nNOSγ within exon 5 generates a truncated 125 kDa isoform. Both nNOSβ and nNOSγ lack the PDZ domain and therefore are localized to the cytosolic fraction.* In vitro assays have shown that nNOSγ lacks significant catalytic activity, whereas nNOSβ possesses activity comparable to nNOSα. nNOS-α, β, and γ are majorly expressed in the brain [85].

**Figure 3 molecules-29-00089-f003:**
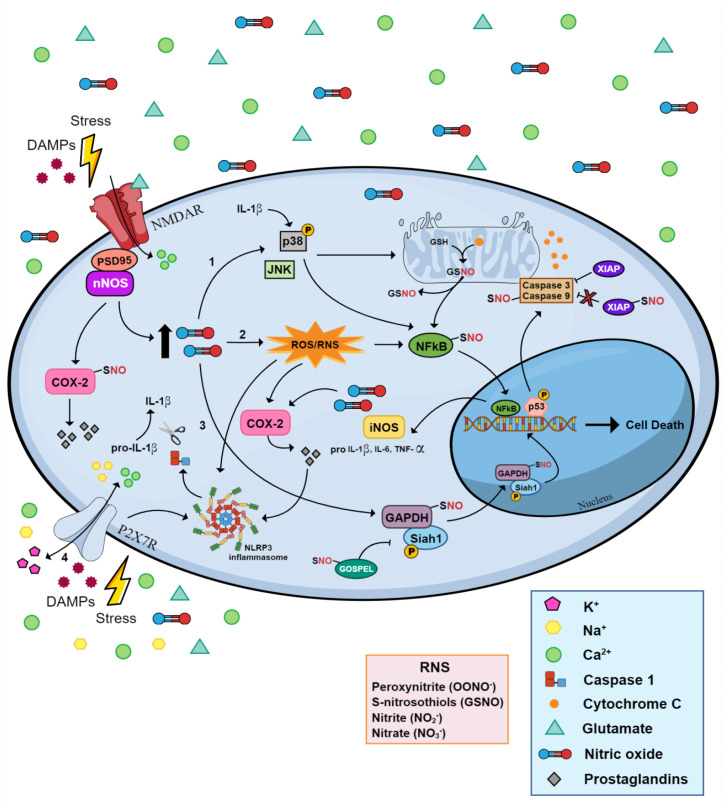
Schematic of the involvement of NO in inflammatory and cell death pathways under stress conditions. In stressful situations, DAMPs like HMGB1 (mainly released by activated microglia and acting via TLR4 or RAGE; not shown) trigger the release of glutamate and the activation of NMDA-type glutamate receptors. These receptors are organized by PSD95, a protein containing multiple PDZ domains. PSD95 also facilitates the coupling of nNOS to NMDAR, enabling the direct activation of nNOS by the influx of Ca^2+^ and subsequent NO production. nNOS, through its PDZ domain, forms a binding interaction with COX-2 and the generated NO S-nitrosylates, ultimately leading to the production of prostaglandins (PGs), resulting in the production of chemokines and cytokines. Following exposure to stress, enhanced NO levels operate through various pathways: (1) Activation of JNK and p38, initiating a pro-apoptotic NO signal and resulting in the phosphorylation and accumulation of p53, activation of caspases, release of cytochrome c, and subsequent apoptosis. Under physiological conditions, XIAP, an E3 ubiquitin ligase, effectively inhibits caspases by targeting them for proteasomal degradation. However, under nitrosative conditions, NO inactivates the XIAP’s E3 ligase activity through S-nitrosylation, leading to caspase stabilization, and sensitization of neurons to apoptotic stimuli. Damaged mitochondria produce GSNO by transferring the nitrosyl group from cytochrome c’s heme iron to GSH. GSNO then translocates to various subcellular locations and participates in transnitrosylation, affecting interacting proteins such as NF-κB. (2) Production of ROS and RNS, responsible for S-nitrosylation of NF-kB, which translocates to the nucleus, leading to the production of iNOS and cytokines (e.g., pro-IL-1β, IL-6, and TNF-α). RNS also activates COX-2, increasing PGs release, and activation of NLRP3 inflammasome. (3) S-nitrosylation of GAPDH at Cys150 enables its interaction with Siah1, which escorts SNO-GAPDH to the nucleus, eliciting cell death. In the cytoplasm, SNO-GAPDH also interacts with GOSPEL, preventing its translocation to the nucleus, thereby inhibiting apoptosis. The competition between SNO-GOSPEL and Siah1 for SNO-GAPDH binding represents a regulatory mechanism that maintains cellular homeostasis in response to stressors. This balance may be disrupted in stress-related disorders such as PTSD. (4) Additionally, stress and DAMPs trigger the efflux of intracellular K+ through P2X7R, ultimately activating the NLRP3 inflammasome complex. This leads to the activation of caspase 1, which cleaves pro-IL-1β into IL-1β. One of the functions of IL-1β is to activate p38.

**Figure 4 molecules-29-00089-f004:**
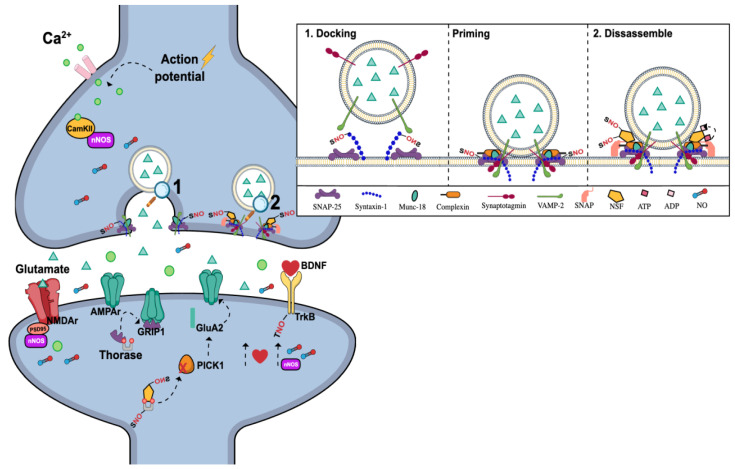
nNOS and the regulation of synaptic transmission and plasticity. After an action potential and influx of Ca^2+^ in the neuron, nNOS is activated through its domain CaM, producing NO. (1) In the docking step, NO can promote the S-nitrosylation of Sec1, inhibiting its binding to Munc-18 and thereby facilitating syntaxin 1 engagement with SNARE and fusion with the membrane. (2) After priming, nNOS might affect the disassembly of the SNARE complex by S-nitrosylating NSF, preventing its ATPase activation by SNAPs. Regarding postsynaptic transmission, S-nitrosylated thorase promotes the stabilization of the GluA2/GRIP1 complex and enhances AMPA endocytosis. On the other hand, thorase can activate NSF by transnitrosylation, promoting NSF binding on GluA2-PICK1, enhancing insertion of GluA2 at the synapse. NO can affect synaptic plasticity through the inhibition of TrkB activation by S-nitrosylation. Concomitantly, BDNF also upregulates the expression of nNOS and augments NO production, indicating that nNOS also plays a role in synaptic plasticity.

**Table 1 molecules-29-00089-t001:** NO modulation affects stress response, and stress exposure influences NO signaling.

Paradigm	Animals	Experimental Condition	Treatment	Main Results	Refs.
NO modu-lation	Male Sprague Dawley rats	TDS stress	7-NI (20 mg/kg; i.p.)	Reversal of ↑ NO_x_ levels in HIP (7 days post-stress)	[16]
	Male iNOS KO mice	-	-	↑ NOS activity in neocortex	[23]
			L-NAME (50 mg/kg; i.p.)	Reversal of anxiety-like behavior in the EPM	
	Male Wistar rats	7-, 14- or 21-days RS (360 min each)	-	↑ NOS activity and NO_x_ after 7 days in cortex; ↑ iNOS expression after 14 days in cortex	[24]
			AG (400 mg/kg; i.p.) from day 7 to 21	Reversal of ↑ NO_x_ levels in cortex	
	Male and female Swiss mice	Acute RS (360 min)	AG (12.5, 25, and 50 mg/kg; i.p.)	Reversal of anxiety-like behavior in the EPM and LDT (50 mg/kg); ↓ plasma nitrite levels	[26]
			SIL (1 mg/kg; i.p.)	Anxiety-like behavior in the EPM and LDT; ↑ plasma nitrite levels	
	Male Wistar rats	Acute RS (120 min)	1400 W (10^−4^, 10^−3^, and 10^−2^ nmol; intra-PL PFC)	Reversal of anxiety-like behavior in the EPM (10^−3^ nmol)	[27]
	Male Sprague Dawley rats	TDS stress	-	↑ NOS activity after the initial stressors and on day 42 in HIP; ↓ GABA levels and NMDA receptor density on day 42 in HIP	[28]
			AG (50 mg/kg; i.p.) from day 1 to 21	Reversal of ↑ NOS activity on day 42 in HIP	
	Male iNOS KO mice	-	-	↑ CFC; ↑ NO_x_ and nNOS expression 24 h after conditioning in medial PFC	[64]
			7-NI (30 mg/kg; i.p.)	Reversal of ↑ CFC response	
	Male Sprague Dawley rats	Acute FSS (1 mA)	L-NAME (50 µL; i.c.v. or 50 mg/kg; s.c.)	Reversal of ↑ plasma ACTH levels (50 μL); ↓ Anterior pituitary NOS activity (50 mg/kg); ↓ Hypothalamic NOS activity	[99]
	Male Wistar rats	Acute RS (180 min)	NPLA (0.04 nmol; intra-PL PFC)	Anxiety-like behavior in the EPM; ↑ nNOS expression after 24 h or 7 days	[100]
	Male Wistar rats	Acute RS (120 min)	L-NAME (10 mg/kg; i.p.)	Reversal of anxiety-like behavior in the EPM; Reversal of depression-like behavior in the FST	[101]
		15 days RS (120 min each)			
	Swiss mice (unk. sex)	Acute or 5-days FSS (0.5 mA or 1.5 mA)	L-NAME (10 and 30 mg/kg; i.p.)	Reversal of ↑ serum CORT levels (1.5 mA; 30 mg/kg)	[102]
			L-arginine (100 and 300 mg/kg; i.p.)	Augment the ↑ serum CORT levels (0.5 mA; 300 mg/kg)	
			L-NAME (30 mg/kg) + L-arginine (300 mg/kg); i.p.	Inhibited L-NAME effects (1.5 mA)	
	Male WT mice	CMS	-	↑ nNOS expression after 4, 21, and 56 days in HIP; ↑ NOx and nNOS activity after 21 days in HIP; Depression-like behavior in the TST	[103]
	Male WT mice		7-NI (30 mg/kg; i.p.) for 7 days	Reversal of depression-like behavior in the TST; ↑ BrdU^+^ cells in the DG	
	Male nNOS KO mice		-	Reversal of depression-like behavior in the TST; ↑ BrdU^+^ cells in the DG	
	Male C57BL/6 mice	CSDS	-	Depression-like behavior in the TST and SPT; ↑ Density of neurons expressing nNOS, nNOS expression and activity in NAc shell	[104]
			L-VNIO (1.5 mM; intra-NAc shell)		
			Carboxy-PTIO (10 µM; intra-NAc shell)	Reversal of depression-like behavior in the TST and SPT	
		CMS	-	Depression-like behavior in the TST and SPT; ↑ nNOS expression in HIP	
	Male WT mice	CMS	-	↑ nNOS expression and density of neurons expressing nNOS in HIP	[105]
			7-NI (10 µM; intra-HIP)		
		CORT-induced chronic stress model	7-NI (10 µM; intra-HIP)	Reversal of depression-like behavior in the TST, FST, and SPT; Reversal of ↑ GR expression in HIP; ↓ CORT plasma levels	
Stress Expos-ure on NO signal-ing	Male Wistar rats	Predator exposure	-	Anxiety-like behavior in the EPM; ↑ NO_x_ and density of neurons expressing nNOS in PFC	[17]
				Anxiety-like behavior in the EPM; ↑ Density of neurons expressing nNOS in BLA	
	Male Wistar rats	Acute RS (360 min)	-	↑ NOS activity after 6 h in cortex; ↑ NF-kB translocation to the nucleus; ↑ iNOS expression in cortex	[25]
	Male Wistar rats	Acute RS (120 min)	-	↑ Density of neurons expressing nNOS in CeA	[106]
		5-days RS (120 min each)	-	↑ Density of neurons expressing nNOS in HIP and entorhinal cortex	
	Male Wistar rats	Acute RS (60, 120, or 240 min)	-	↑ CORT plasma levels, NO_x_ and constitutive NOS activity after 60 min in HIP; ↑ *iNOS* gene expression and activity after 240 min in HIP	[107]
				↑ *iNOS* gene expression and activity after 120 min in striatum	
	Male Wistar rats	Acute RS (60, 120, or 240 min)	-	↑ *nNOS* and *iNOS* gene expression after 240 min in PFC	[108]

Abbreviations: ↑: increased; ↓: decreased; 7-NI: 7-nitroindazole; ACTH: adrenocorticotropic hormone; AG: aminoguanidine; BLA: basolateral amygdala; CeA: central amygdala; CFC: contextual fear conditioning; CMS: chronic mild stress; CORT: corticosterone; CSDS: chronic social defeat stress; DG: dentate gyrus; EPM: elevated plus-maze; FSS: foot shock stress; FST: forced swimming test; GR: glucocorticoid receptor; HIP: hippocampus; i.c.v.: intracerebro ventricular; iNOS: inducible nitric oxide synthase; i.p.: intraperitoneal; KO: knockout; LDT: light and dark test; L-NAME: L-NG-Nitro arginine methyl ester; L-VNIO: N5-(1-imino-3-butenyl)-L-ornithine; NAc: nucleus accumbens; nNOS: neuronal nitric oxide synthase; NO: nitric oxide; NO_x_: total nitrite and nitrate; NPLA: N-propyl-L-arginine; PFC: prefrontal cortex; PL: prelimbic; RS: restrain stress; s.c.: subcutaneous; SIL: Sildenafil; SPT: sucrose preference test; TDS: time-dependent sensitization; TST: tail suspension test; unk.: unknown; WT: wild-type.

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
