# Peer review of "“NO” Time in Fear Response: Possible Implication of Nitric-Oxide-Related Mechanisms in PTSD"

_molecules, 2023, doi:10.3390/molecules29010089_

Round 1

Reviewer 1 Report

Comments and Suggestions for Authors

I find the review of excellent quality, the contribution given to this field of research is really appreciable because it defines and collects knowledge about the role of NO in psycho-behavioral disorders.

However the amount of information collected is very extensive, my advice is to try to insert more diagrams and figures, not only on the way of reporting NO but also on the studies mentioned in the work, for example clinical and pre-clinical evidence.

whether the authors are aware of this, are there any hypotheses or attempts to apply this knowledge therapeutically? in this case include this part in the conclusions

Author Response

Reviewer 1:  “I find the review of excellent quality, the contribution given to this field of research is really appreciable because it defines and collects knowledge about the role of NO in psycho-behavioral disorders.”

Point 1. However, the amount of information collected is very extensive, my advice is to try to insert more diagrams and figures, not only on the way of reporting NO but also on the studies mentioned in the work, for example, clinical and pre-clinical evidence.

Author's reply: Thank you, we genuinely appreciate your suggestion. This version of the manuscript contains now 3 Figures and a graphical abstract. In the Figures, we tried to address major important aspects of NO, including the variant transcripts of nNOS mRNA (Figure 1), involvement of NO in inflammatory and cell death pathways (Figure 2) and how NO derived from nNOS regulates synaptic transmission and plasticity (Figure 3). We also added a Table summarizing the main findings regarding the effect of NO modulation in behavioral consequences of stress exposure and the impacts of stress exposure on NO signaling (Table 1). We hope we have improved the clarity of the manuscript and attended to your request.

Point 2. Are there any hypotheses or attempts to apply this knowledge therapeutically? in this case include this part in the conclusions.

Author's reply:  This is indeed an excellent point, thank you. There are no ongoing or ended clinical trials addressing the therapeutic potential of NOS inhibitors for any fear, anxiety, or mood-related disorders. However, in the neurology field some inhibitors have been tested, including for brain ischemia, traumatic brain injury, and sepsis. Based on this, we modified the conclusion adding a reference for that matter (Page 34):

NO has been extensively associated with fear-related mechanisms, particularly through the activity of nNOS. However, its regulatory influence extends beyond simple NOS activation and NO production, encompassing various cellular mechanisms and pathways. In this sense, this review explored the intrinsic role of NO and its many associated pathways, especially in conjunction with nNOS, in the regulation of fear responses, stress, and their implications in pathophysiological alterations seen in PTSD. By unraveling the complexities of nNOS signaling, this study opens new avenues for understanding the profound impact of NO and NO-induced PTMs on the mechanisms affecting fear memory circuitry, offering potential therapeutic insights into fear-related disorders. These mechanisms, ranging from stress response and neuroinflammation to oxidative stress, cell death, and synaptic transmission/plasticity, constitute a crucial aspect of the physiological underpinnings of PTSD and related conditions.

Although, from our knowledge, no drug directly targeting NOS isoforms has been investigated in clinical trials for fear or anxiety-related disorders, one NOS inhibitor (Ronopterin - VAS203) and one Discs large homolog 4 inhibitor/NMDAR antagonist (Nerinetide - NA1 tat-NR2B9c) are currently being tested in Phase 3 clinical trial for traumatic brain injury (TBI) and/or ischemia [285]. If the efficacy and safety of these drugs are confirmed in these trials, they could open the possibility of testing them in other conditions, including PTSD.

Reviewer 2 Report

Comments and Suggestions for Authors

In this review, Fronza and coworkers discuss the intrinsic role of NO and its many associated pathways, particularly involving nNOS, in the regulation of fear responses, stress, and their implications in pathophysiological alterations in PTSD. The review describes thoroughly NO-related molecular mechanisms, mainly posttranslational modifications (PTMs), the role of NO in cell death, metabolism, stress and the potential interaction of these mechanisms within the brain's fear circuitry.

The review is well written and interesting. Nitrosative stress and S-nitrosylation is rather interesting topic in stress-related pathologies.

I have only minor comments.

-      I find the description of each molecular cascade extremely complete and detailed, however for some of the most emphasized molecular targets (BDNF, AMPA, DAMPs, SNARE. NMDA..) the link between NO and PTSD remains elusive.

-      In the title they state: possible implication of inhibiting nitric-oxide-related mechanisms in PTSD, is not clear how they address this part in the text. The authors report the several results in which NO-modulators, nNOs inhibitors, NO-donors, 8-nitro cGMP, or others compounds counteract the reported effects in memory formation or fear acquisition but the real potentiality for PTSD and its cure does not stand out.

-      The Authors should also address that fear is just one of the behavioural phenotypes of PTSD

-      The only preclinical model quoted is the single prolonged stress (SPS), otherwise the authors refers to generical animal model of PTSD (line 46, 60, 460, 596), given the variety of the animal models of PTDS. the authors should specify what models they are referring to and possibly include data from other models (if available) for examples the following studies seems on topic: PMID: 35605060, PMID: 32083948, PMID: 30125872.

-      Wang et al. (2018) not reported in the reference list.

Author Response

Reviewer 2. In this review, Fronza and coworkers discuss the intrinsic role of NO and its many associated pathways, particularly involving nNOS, in the regulation of fear responses, stress, and their implications in pathophysiological alterations in PTSD. The review describes thoroughly NO-related molecular mechanisms, mainly posttranslational modifications (PTMs), the role of NO in cell death, metabolism, stress and the potential interaction of these mechanisms within the brain's fear circuitry. The review is well written and interesting. Nitrosative stress and S-nitrosylation is rather interesting topic in stress-related pathologies. I have only minor comments.

Point 1. I find the description of each molecular cascade extremely complete and detailed, however for some of the most emphasized molecular targets (BDNF, AMPA, DAMPs, SNARE. NMDA..) the link between NO and PTSD remains elusive.

Author reply. Thank you for the note. We have added some clinical evidence indirectly linking AMPA and NMDA to PTSD, as well as the neurotransmitter exocytosis machinery  in the page 27, last paragraph, page 28, paragraph 1, as you can see in the following paragraph:

Changes in functional connectivity and synaptic transmission have been suggested to be an important feature of PTSD pathophysiology. For example, reduced prefrontal glutamatergic synaptic strength [239] and higher cortical metabotropic glutamate receptor 5 (mGluR5) have been described in PTSD patients [240];. Also, these patients show higher levels of glutamate and lower levels of the integrity neuronal marker N-acetyl aspartate (NAA) [194]. Regarding the genetic predisposition, a variation in a glutamate transporter gene (SLC1A1), which codes for the excitatory amino acid transporter 3 (EAAT3) and excitatory amino acid carrier 1 (EAAC1), has been associated with the risk for developing PTSD and greater symptom severity [241]. These pieces of evidence support the proposal that glutamate synaptic changes and altered glutamate availability are crucial events in trauma and stress-related psychopathologies.

Regarding to brain plasticity/BDNF we added the following sentence in the page 33, paragraph 2: 

Compelling evidence suggests that NO, primarily produced by nNOS, leads to the downregulation of CREB transcriptional activity [273]. This downregulation, which exhibits an anti-proliferative effect, has been observed in both the hippocampal dentate gyrus and the subventricular zone [274,275]. It is interesting that PTSD patients showed reduced hippocampal volume relative to control subjects [276]. Typically, CREB is activated through Ser133 phosphorylation in response to external signals, such as those from neurotrophins or neurotransmitters [277]. Interestingly, CREB activation by BDNF appears to be modulated by a NO-dependent signaling pathway involving S-nitrosylation of components within the CREB DNA-binding complex [278]. In this context, although the NO/GAPDH/Siah1 signaling pathway plays a role in cell death, it also promotes CREB-mediated dendrite outgrowth through epigenetic chromatin-remodeling regulatory mechanisms [279], indicating the broad effects mediated by NO.

Point 2.  In the title they state: “Possible implication of inhibiting nitric-oxide-related mechanisms in PTSD”, is not clear how they address this part in the text. The authors report the several results in which NO-modulators, nNOs inhibitors, NO-donors, 8-nitro cGMP, or others compounds counteract the reported effects in memory formation or fear acquisition but the real potentiality for PTSD and its cure does not stand out.

Author reply.  Thank you for pointing that out. Considering the lack of clinical evidence, we changed the title to:“NO” time in fear response: possible implication of nitric-oxide-related mechanisms in PTSD.

Point 3. The Authors should also address that fear is just one of the behavioural phenotypes of PTSD.

Author reply. We apologize for not having made that clear. We added information about that to the sentence in the page 2 paragraph 3: The manifestation of exaggerated and abnormal fear responses, particularly following traumatic experiences, represent a distinct behavioral phenotype within the spectrum of PTSD. This behavioral profile is characterized by the elicitation of intense feelings of fear, horror, or helplessness. Additionally, PTSD is marked by the pervasive presence of intrusive memories, avoidance behaviors and persistent negative changes in thinking and mood  [3,4].

Point 4. The only preclinical model quoted is the single prolonged stress (SPS), otherwise the authors refers to generical “animal model of PTSD” (line 46, 60, 460, 596), given the variety of the animal models of PTDS. The authors should specify what models they are referring to and possibly include data from other models (if available) for example, the following studies seems on topic: PMID: 35605060, PMID: 32083948, PMID: 30125872.

Author reply. Thank you. We hope that the Table 1 will give more clarity to this point of the text. We named the stressors as immobilization, cat odor exposure, and time-dependent sensitization as models of PTSD. We also added in the text the references: chronic isolation stress (PMID: 35605060) and the electric shock stress (PMID: 32083948) in the following paragraphs: 

Page 6, topic 2:  NO involvement in learned fear

nNOS is also important for fear extinction. Administration of the nNOS inhibitor ZL006, a drug that uncouples nNOS from PSD95, into the CA3 of the hippocampus facilitates fear extinction [70], and its administration into the dentate gyrus promotes extinction retrieval of remote fear extinction [71]. Corroborating with this data, the combination of MK‐801 (NMDA antagonist) and L‐NNA (NOS inhibitor) treatment ameliorated the extinction learning deficits in mice exposed to chronic social isolation stress [72]. Moreover, iNOS KO mice, which present increased NOS activity in the prefrontal cortex and altered expression of nNOS and eNOS enzymes, also have deficits in fear extinction which were attenuated by systemically 7-NI [64]. Interestingly, ZL006 in the anterior cingulate cortex (ACC) also inhibited contextual fear generalization in a novel context [73], another feature of PTSD.

Page 14, topic 4. nNOS, NO-related mechanisms, and stress response

Corroborating with a possible causal role for increased NO in the brain in behavioral modifications after stress exposure, the anxiogenic effect induced 24h after an acute 2h-restrain stress session, in view of modeling a trauma, was correlated with increased nNOS levels in the PL PFC. Moreover, the anxiogenic behavior was prevented by the administration of an nNOS inhibitor (N-propyl-L-Arginine; NPLA) in the same brain region [100]. Besides, the administration of L-arginine or L-NAME in the basolateral amygdala (BLA) prevented the long-term alterations in anxiety and depressive-like behavior induced by chronic stress [101].  Furthermore, adding to the complexity of NO and stress response, the administration of L-arginine (300 mg/kg) abolished the stress adaptive response in mice subjected to a 5-day of low-intensity foot-shock stress protocol, while L-NAME (30 mg/kg) favored stress adaptation in mice submitted to a high-intensity foot-shock stress [102]. The nNOS overexpression, at least in the hippocampus of mice exposed to chronic mild stress, is crucial for the development of stress-induced depressive behavior [103], suggesting the involvement of nNOS in modulating coping behavior in response to stress. In line with this data, the nNOS in the nucleus accumbens, a reward center, regulates susceptibility to social defeat stress and subsequent depression-like behaviors in mice [104].

Point 5. Wang et al. (2018) not reported in the reference list.

Author reply. Thank you. The reference was included.